# Accounting for the Biological Complexity of Pathogenic Fungi in Phylogenetic Dating

**DOI:** 10.3390/jof7080661

**Published:** 2021-08-14

**Authors:** Hannah M. Edwards, Johanna Rhodes

**Affiliations:** MRC Centre for Global Infectious Disease Analysis, Imperial College School of Public Health, Imperial College London, Norfolk Place, London W2 1PG, UK; johanna.rhodes@imperial.ac.uk

**Keywords:** evolution, methodology, mutation rates, phylogenomics, recombination

## Abstract

In the study of pathogen evolution, temporal dating of phylogenies provides information on when species and lineages may have diverged in the past. When combined with spatial and epidemiological data in phylodynamic models, these dated phylogenies can also help infer where and when outbreaks occurred, how pathogens may have spread to new geographic locations and/or niches, and how virulence or drug resistance has developed over time. Although widely applied to viruses and, increasingly, to bacterial pathogen outbreaks, phylogenetic dating is yet to be widely used in the study of pathogenic fungi. Fungi are complex organisms with several biological processes that could present issues with appropriate inference of phylogenies, clock rates, and divergence times, including high levels of recombination and slower mutation rates although with potentially high levels of mutation rate variation. Here, we discuss some of the key methodological challenges in accurate phylogeny reconstruction for fungi in the context of the temporal analyses conducted to date and make recommendations for future dating studies to aid development of a best practices roadmap in light of the increasing threat of fungal outbreaks and antifungal drug resistance worldwide.

## 1. Introduction

Phylogenetic inference can be a powerful tool to understand the evolutionary history of a pathogen, including species divergence, time to most recent common ancestors and, combined with spatial data, when it may have emerged in a new geographic region. When phylodynamic models are also applied, certain processes that have shaped this evolution can be inferred, such as temporal shifts in selection pressure, changes in effective population size, and spatial dynamics of the host population. Insights can therefore be gathered on how the epidemiology and biology of pathogens have been shaped by this ecological and evolutionary history [1,2].

Scaling of divergence times in phylogenetics makes use of estimates of molecular clock rates. Sampling dates are used to fix the ages of the tips in the phylogenetic tree and estimation of molecular clock rates enables dating of internal nodes corresponding to coalescent or birth events where lineages diverged from their common ancestor [3,4]. The ability to do this is dependent on how the rates of evolutionary and ecological changes relate to each other and there is a requirement for a high amount of detectable genetic variation over the time scale from which the genomes being analysed have been isolated [5]. Hence, in the study of microbial pathogens, these analyses have been limited largely to viruses and some bacteria whose rate of evolutionary change can be detectable over the course of an outbreak. For example, estimates of the mutation rate of ebolavirus from the 2014 West African epidemic suggest a mutation rate of ~1.2 × 10^−3^ [1.13, 1.27 × 10^−3^] nucleotide substitutions (subs) per site per year [6], while Zika virus has been estimated to range from 0.6 × 10^−3^ [0.3, 0.8 × 10^−3^] to 0.8 × 10^−3^ [0.5, 1.1 × 10^−3^] subs/site/year [7]. Temporal analyses have been applied less to pathogenic fungi, which typically evolve much slower, although some do evolve within the same range of bacteria to which dating analyses have been applied more frequently. Bacterial mutation rates span several orders of magnitude from ~10^−5^ to 10^−8^ subs/site/year [8] including one of the most commonly analysed bacteria, *Mycobacterium tuberculosis* (MTB), which evolves an estimated 0.5 subs/genome/year, or 1.1 × 10^−7^ subs/site/year given its 4.4 Mb genome [9]. In comparison, our review of the published literature identified primary dating analysis in just four studies on pathogenic fungi, including *Cryptococcus gattii* [10], *Paracoccidioides* spp. [11], and *Candida auris* [12,13], estimating mutation rates in the order of 10^−8^, 10^−9^, and 10^−5^ subs/site/year for the three pathogens, respectively. It is worth noting, however, that these four species do not represent all fungal phyla with pathogenic members.

Despite limited focus so far, application of temporal analyses such as these is increasingly relevant given fungal outbreaks are increasing in frequency and scope [14,15,16,17,18]. Scaling of divergence times in a phylogeny from genetic distance into absolute units of time can aid in the investigation of pathogen outbreaks—including timing when and from where an outbreak occurred and how it spread—transmission events, and spread of drug resistant clades [2,5,19,20]. Population dynamics can also be timed with historic events, such as human migrations [21], political events [22,23], or public health control measures [24,25]. The yeast *C. auris*, which causes an invasive blood stream infection, has been the cause of several recent nosocomial outbreaks in contaminated settings with subsequent transmission between patients [26,27,28]. Accurately recreating emergence and transmission events can help identify likely sources of introduction and how such hospital outbreaks have occurred. Several pathogenic fungi have also shown increased resistance to frontline drugs, including *Aspergillus fumigatus* [29], *C. auris* [30,31], and *Cryptococcus neoformans* [32,33], while others have shown recent emergence in global regions previously free of the pathogen and subsequent disease, for example, the emergence of *C. gattii* in the Pacific Northwest [34,35]. Being able to understand and date these evolutionary histories is key to understanding the processes that shape resistance emergence and spread to new global populations and patient cohorts.

Fungi are complex eukaryotic organisms, with several biological processes that may hinder accurate phylogenetic reconstruction and temporal dating. Fungal mutation rates are slower than those of viruses and (most) bacteria, which may impede detection of a temporal signal among collected isolates, and they have complex reproductive processes leading to potentially high levels of recombination and genetic variation. Fungi also display rate heterogeneity. Multiple species have been shown to display hypermutation, an excess of mutations throughout the genome, which would artificially inflate any clock rate discerned, as well as the same species entering dormant states where mutation rates may be reduced. Here, we discuss the methodological implications of these aspects of fungal biology on temporal analyses, and how they can be addressed in the context of the four primary pathogenic fungal dating studies we have identified to date (herein referred to by their first author: Chow [13], Rhodes [12], Roe [10] and Munoz [11]). These studies utilise differing methodologies with various strengths and deficits to each approach. We end with some recommendations to guide methodological choices in future temporal studies on pathogenic fungi.

## 2. Methodological Considerations for Temporal Analyses of Pathogenic Fungi

### 2.1. Detection of a Temporal Signal in Sequence Data

Viruses and bacteria lend themselves well to phylogenetic inference because of their relatively high mutation rates, in the order of 10^−^^8^ to 10^−^^6^ substitutions per nucleotide site per cell infection (s/n/c) for DNA viruses and 10^−^^6^ to 10^−^^4^ s/n/c or higher for RNA viruses, and 10^−^^5^ to 10^−^^8^ s/site/year in bacteria [8,36,37,38]. However, these can often be higher among outbreak lineages (see rate variation discussion below). Therefore, over the course of an outbreak, isolates often show enough genetic variation to enable estimation of the molecular clock. The presence of temporal signal in a set of isolates is an essential pre-requisite for subsequent phylogenetic dating analysis. Sampled isolates have to be spread over an adequate time frame such that the substitution rate can be accurately estimated as there has to be a measurable amount of evolutionary change [39].

Temporal signal can be assessed via a linear regression of sampling time (date of isolation) against root-to-tip distance in the tree (e.g., number of nucleotide substitutions), where the slope corresponds to the substitution rate assuming a strict molecular clock, and where R^2^ is a measure of how much the evolution shows a temporal signal [40] (Figure 1). This on its own, however, does not assess whether the signal is significantly different to what may be observed by chance. An alternative/additional method to determine whether there is a clock-like signal is to compare models with and without the inclusion of sampling dates. If the addition of sampling dates improves the model fit, then this may be indicative of temporal signal in the data [41,42]. These standard tests for temporal signal can still be insufficient, though. The rates and timescale can be spurious if the number of samples is small, if the sampling period is too short, or if closely related sequences are more likely to have been sampled at similar times [41].

Another method of validating rates and timescales is to use a date-randomisation test using replicate datasets in which sampling times have been randomised [43]. If the estimated mean substitution rate of the data in question lies outside of the 95% credible intervals of the rate estimates calculated from the replicate datasets, then it is indicative of a significant temporal signal. In tests, however, this can fail to detect rate estimates from data with no temporal signal, and it can also perform poorly when sampling times are not uniformly distributed through the tree [44]. As such, a more conservative criterion is recommended, whereby the 95% credible intervals of both the mean estimate from the data and the estimates provided by the randomised replicates do not overlap [44]. Despite this, the randomised permutation approach may still be inappropriate if the genetic and temporal data are confounded such as when closely related sequences are sampled at the same time. In this case, a clustered permutation approach has been proposed [41,44].

Fungi tend to mutate at slower rates than viruses and bacteria, often in the range of 10^−8^ to 10^−^^10^ substitutions per base pair per generation [45,46]. This may affect the ability to detect a temporal signal, particularly if the isolates being analysed have been collected over a short timeframe. Of the four studies on pathogenic fungi reviewed here, three tested their data for evidence of temporal signal (Roe, Rhodes, and Chow), and one (Munoz) did not. *C*. *auris* is an intrinsically drug-resistant nosocomial pathogen. Population genomic studies have identified a strong phylogenetic structure containing four clades representing geographical regions: South Asia (Clade I), East Asia (Clade II), Africa (Clade III), and South America (Clade IV). Each Clade is separated by tens of thousands of single-nucleotide polymorphisms (SNPs). The studies investigating *C. auris* (Chow and Rhodes) each found mutation rates in the order of 10^−^^5^ for their *C. auris* clades, a higher mutation rate than is typical of fungi, and thus perhaps enabling detection of a temporal signal, despite a narrow window of time within which isolates were collected. However, each study only conducted linear regression analyses to determine the temporal signal. Chow identified R^2^ values of 0.55, 0.56, and 0.21 for *C. auris* Clades I, III, and IV, respectively, while Rhodes found an R^2^ of 0.37 for Clade I. In each instance, these were assessed to be indicative of sufficient clock-like signal to justify subsequent dating and did not do further analysis to prove the significance of the temporal signal.

Roe’s study on *C. gattii* also utilised regression analysis but went further to determine the statistical significance of these measures by conducting 10,000 random date permutations for each of the clades. For the three sub-populations of *C. gattii* analysed, they identified R^2^ values of 0.5971, 0.661, and 0.0745 for clades VGIIa, VGIIb, and VGIIc, respectively, indicative of a strong clock-like signal for VGIIa and VGIIb and weak signal for VGIIc. However, the random permutations did not find any of these to be significant from chance at the 0.05 confidence level, with derived *p*-values of 0.076, 0.268, and 0.294, respectively. Despite the lack of evidence for a statistically significant temporal signal in the data, the authors continued with their dating analysis, and this has been adopted to theorise about how and when *C. gattii* may have emerged in the Pacific Northwest (PNW) [47].

It should be a priority for future fungal dating studies to rigorously test for a temporal signal in their data and employ statistically significant cut-offs for these signals. Erroneous assumption of a temporal signal would lead to incorrect estimation of the molecular clock rate and incorrect dating of the phylogeny.

### 2.2. Recombination Needs to Be Considered and Accounted for

Pathogenic fungi, such as *Cryptococcus* and *Candida*, propagate predominantly via asexual budding, whereby haploid cells undergo mitosis to produce clonal haploid progeny [48]. However, these fungi also occur as one of two mating types that can reproduce sexually (albeit infrequently), generating novel phenotypes [49,50]. Populations of pathogenic fungi involved in outbreaks generally appear highly clonal; however, these outbreak lineages typically stem from environmental populations with a much greater pool of diversity where evidence of sexual reproduction can be found. This ability to switch between reproductive modes complicates the fungal life cycle and allows for the emergence of new and, occasionally, more virulent genotypes. For example, a clonal population of *C. gattii* VGIIa was the major lineage involved in the Vancouver and PNW outbreak of cryptococcosis and emerged with a hypervirulent phenotype from less virulent progenitors through a combination of microevolution, recombination, and sexual reproduction [50].

Fungal sexual reproduction can be heterothallic when two opposing mating types come into contact and mate with each other, or homothallic, whereby two cells of a single mating type mate with each other [51]. This ability to undergo unisexual reproduction is important because heterothallic reproduction is likely very rare. In *Cryptococcus* spp., for example, the MATα mating type is highly prevalent, but the MATa mating type is rarely found in the environment [52,53], while in *Candida albicans*, heterothallic reproduction is infrequent because it is regulated by the phenotypic switching of MATα or MATa cells into a form that mates more efficiently [54,55]. The same phenotypic switching is not a prerequisite for homothallic mating. Although offspring from unisexual crosses have lower average recombination rates than those derived from bisexual crosses, they do result in divergent and aneuploid progeny able to undergo subsequent meiosis and sporulation [56]. Further complicating the matter, *Saccharomyces cerevisiae*, for example, can undergo rare matings, which are difficult to distinguish from homothallic mating. These rare matings occur when diploid cells heterozygous for mating type and which thus lack mating ability (these cells are referred to as “non-maters”) convert to a homozygous mating type and cross with maters to produce polyploid progeny [57]. Sexual reproduction of diploid *C. albicans* cells is also followed by a non-meiotic process of depolyploidization known as concerted chromosome loss (CCL) to convert tetraploid cell products to a diploid state. This CCL is associated with a three-fold greater rate of recombination than normal mitotic growth [58].

These reproductive states present an issue for temporal analysis through the introduction of potentially high amounts of recombination. Recombination has been shown to disrupt phylogenetic inference through loss of any temporal signal and overestimation of substitution rate heterogeneity, leading to loss of the molecular clock, incorrect branch lengths, and erroneous TMRCA estimates (Figure 2) [42,59,60]. The estimated tree topology can be unreliable and lead to a false inference of positive selection [61,62,63]. Ancestral trees become more “star-like” with longer terminal branches, resembling a phylogeny for which a population is undergoing exponential growth (Figure 3). Lengthened branches analogous to an exponentially growing population occur because recombination makes distances between sequences more similar to each other, although this is dependent on the relatedness of the sequences involved in the recombination event and on the time when the event occurred [59,63,64]. Furthermore, ancestral sequence reconstruction is biased by recombination and can be quite distinct from the actual most recent common ancestor (MRCA), instead resembling a concatenate of partial MRCAs at each recombination fragment [65].

Generally, removal of recombinant regions is advised prior to phylogeny estimation. Comparison of Bayesian dating algorithms between trees uncorrected and corrected for recombination has shown that correcting the tree will identify significant temporal signal between isolates otherwise lost in uncorrected trees and will greatly reduce uncertainty in node and root dates [42]. Despite this, the dating papers we identified for pathogenic fungi have failed to adequately account for recombination in their data. In Chow’s study of *C. auris*, the authors found no evidence of admixture between the four major clades by principal component analysis and genome-wide *F*_ST_ despite each clade being entirely made up of one mating type, potentially allowing for heterothallic reproduction between clades. However, no test for recombination was applied prior to their temporal dating analysis to confirm this, nor to detect recombination sites from reproduction within clades, the latter of which may be more likely since this does not require phenotypic switching. Rhodes identified one isolate that showed less genetic divergence from the root than expected in their root-to-tip divergence analysis, which the authors propose to be possibly due to excessive recombination in this isolate. They did not, however, test isolates for recombination prior to conducting this temporal analysis.

Roe did not find any evidence of recombination between isolates within each of their three sub-population clades of *C. gattii* (VGIIa, VGIIb, and VGIIc); however, they did not consider potential recombination between these clades. These *C. gattii* lineages are highly clonal and isolates have been exclusively of one mating type; however, unisexual mating may occur. Finally, Munoz found evidence of recombination between lineages of *P. brasiliensis*, but they did not account for this in their phylogenetic analysis.

Future studies should seek to adequately consider and identify recombination within their data and remove any identified areas from analysis as appropriate. Tools such as Gubbins [66] and ClonalFrameML [67] offer computationally quick methods with which to identify areas of recombination and accurately reconstruct phylogenies from whole genome sequence data. Gubbins can be applied to a variety of haploid genotypes and large datasets of closely related sequences, although accuracy can reduce when recombination events involve only a small number of bases (the default threshold is set to 3 bp) or when divergence between sequenced isolates increases. Despite these limitations, small recombination events are unlikely to have a large impact on the structure of the inferred genealogy, and the effect of greater sequence divergence can be mitigated, to some extent, by denser sampling or subdivision of the population prior to analysis [66]. These two tools identify recent recombination events between closely related isolates only. A newer tool, fastGEAR, can detect both recent and ancestral recombinations among species-wide gene alignments; however, it does not build a phylogeny [68]. Choice of the correct tool may, therefore, depend on the data and question at hand. Furthermore, these tools have been developed for haploid bacterial genomes and their application to fungi appears largely non-existent in the published literature. A comparison of software tools relevant for dating phylogenies is presented in Table 1. This list is by no means exhaustive but suggests some of the more relevant and popular tools in use. Additional tools relevant to some of the tasks can be found online at: http://methodspopgen.com (accessed on 9 August 2021) [69].

### 2.3. Quiescence to Hypermutation—Potential Impact of Mutation Rate Variation

Similar to some pathogenic bacteria, fungi can show enormous variation in their metabolic and reproductive rates when subjected to external stressors, resulting in slower or faster rates of mutation over time. Best understood and observed is the emergence of hypermutative states in fungi, such as *Cryptococcus* and *Candida*, during infection and exposure to the host environment and anti-fungal drug treatment, leading to emergence of resistant isolates and persistent and/or recurrent infection [45,70,71,72,73,74]. Similar phenomenon is seen in some bacteria, such as *Salmonella enterica* and *Escherichia coli*, which have up to a 1000-fold increase in point mutations in clinical isolates [75,76].

Being difficult to observe in the lab, the process whereby fungal cells enter states of latency, dormancy, and/or quiescence is less well understood [77]. A period of latent infection is suspected in many cryptococcal cases, where infection may occur months-to-years before reactivation and manifestation of disease [78,79,80]. The exact length of this period is currently unknown but may be up to 10 years or more in some cases [81]. The dimorphic fungus *Talaromyces marneffei* (formerly known as *Penicillium marneffei*) can cause a latent infection spanning several decades, reactivating if the patient later becomes immunosuppressed [82]. A similar process is observed in TB, where MTB cells lie dormant in lung and extrapulmonary tissues [83]. Latent tuberculosis (TB) was traditionally thought to comprise a population of nonreplicating and metabolically inactive cells, but there has since been conflicting evidence about whether this is true or not [83]. Some studies have suggested replication continues as normal in latent TB cell populations [84], while others suggest there is a significant slowing down [85,86,87]. A more recent study investigating index TB cases and paired household contacts that developed active TB up to 5.25 years later has suggested a decline in mutation rates 1–2 years after transmission [88].

As well as these latent states of infection, it has been hypothesised that environmental fungi may enter long periods of dormancy (defined by an absence of metabolic activity such as that observed in fungal spores) or quiescence (where metabolic activity is ongoing and cells are able to return to the “normal” cell cycle) in the environment when resources may be scarce [89,90,91,92]. For example, *Saccharomyces cerevisiae* has been shown to enter a dormant state when there is lack of sufficient nitrogen or carbon in its substrate [90]. Previous phylogenetic analysis of cryptococcal isolates in Vietnam showed long terminal branches which could indicate a period of cessation of recombination reflective of a possible quiescent state [93]. It is unknown how often these periods of quiescence occur and how long they last, and it may be difficult to determine their impact on dating estimates. Whereas hypermutation is likely to elevate the molecular clock estimate, one could assume that quiescent and dormant states would make clock rates appear slower than the “normal” rate in these isolates. Isolates under study are most often taken from active infection, and thus, if the potentially elevated clock rate is applied across the whole tree, this could estimate incorrect branching times if the internal branches include latent/dormant states where the mutation rate should be much slower. The impact of quiescence could be more complex, given that this state has been associated with its own “replication-independent” mutational profile in response to external stress in laboratory settings, with more deletions relative to insertions and a reduction in A/T composition—features, which are the antithesis to normal cell cycling conditions [94].

If variation in mutation rates is frequent, then it is unlikely a reliable clock rate can be estimated. Temporal heterogeneity has previously been highlighted as a potential issue for both bacteria and viruses since it may not be represented using current clock models and may even obscure the signal of measurable evolution [95]. At the very least, the presence of such states suggests a relaxed clock model to be optimal to take into account at least some variation in the rate of molecular evolution between lineages and allow non-clock-like relationships between sequences within a phylogeny [96,97,98].

Depending on the data, variations of the relaxed clock model, such as the fixed local clock, whereby the underlying tree has a constant rate, but the rate of each clade is able to differ from this global rate [99]; the uncorrelated relaxed clock, whereby each branch of the tree is allowed its own mutation rate independent of the rate on neighbouring branches [98]; or the random local clock, which forms an interim between the strict and relaxed clocks, where each branch can take a different rate or remain the same as one another [100]. More recently, additive clock models have been proposed that satisfy the additive property often unsatisfied by conventional relaxed clock models [101] and may in most cases be more appropriate than the conventional relaxed clock. In the fungal studies identified, both *C. auris* papers (Chow and Rhodes) utilised strict clock models, seemingly without comparison with alternative options. In their study on *Paracoccidioides*, Munoz used a relaxed clock/uncorrelated lognormal clock for genome-wide variable sites and strict clock for nonpartitioned variant sites. Finally, Roe tested a combination of relaxed lognormal and strict clock models and chose the best fitting model for each clade, which were strict clock models for VGIIb and VGIIc and a relaxed lognormal model for VGIIa.

More work is needed to understand when and how often fungi may enter these different states and what effect this may have on dating estimates. In the meantime, any analyses should be interpreted with caution and alternative clock models considered.

### 2.4. Potential Sensitivity to Priors Requires Explicit Sharing of Bayesian Model Settings

A widely recommended tool for phylogenetic and dating analysis is BEAST (Bayesian estimation analysis for sampling trees) which uses Bayesian Markov chain Monte Carlo (MCMC) models to estimate phylogenies and the associated parameters, including effective population size, TMRCAs, and clock rates [102]. Although many studies have suggested Bayesian analysis may be more powerful than comparative maximum likelihood methods, they can be complicated to implement since they require a high amount of computational power as well as knowledge of prior information to inform suitable model parameters, including priors on the tree model, clock rate, population size, substitution model, and partitioning of variant and non-variant sites [103]. Over-parameterisation of the models can result in inference difficulties and/or computational problems, while under-parameterization can lead to incorrect phylogenetic trees and biased estimates of branch lengths and substitution rates with spuriously high posterior probabilities and over-confident uncertainty estimates [104].

Setting correct priors is essential as certain estimates can be sensitive to the priors placed on them [105]. TB, for example, has been shown to be sensitive to both the clock prior and tree prior if the temporal signal is weak in the data set [37]. The approach to partitioning of substitution rates can also affect tree topology, branch lengths and bootstrap support, and divergence time estimation [106,107]. As such, as well as robustly measuring for temporal signal (as described above), it is essential to provide information on the priors set in such analyses, so that any biases can be identified and ensure findings can be replicated and updated as necessary. For this reason, publication of the XML files used for any BEAST analysis is strongly encouraged although appears to be rarely done in practice [102]. In our identified studies, three made use of Bayesian dating analysis using BEAST. None of these studies provides the full model details or XML file used, and the level of detail in the relevant Methods section varies. Furthermore, it is unclear whether two (Munoz and Roe) of the studies removed burn-in from the MCMC algorithm, even though this is a necessary methodological step.

## 3. Recommendations for the Way Forward

There is no doubt that with the increasing incidence of fungal outbreaks, spread of anti-fungal drug resistance, and appearance of fungal strains in new geographic locations, temporal dating analyses, as applied in phylodynamic models, will be an invaluable tool to understanding these processes and helping to curb the health impact. Although generally slower, fungal mutation rates do overlap with those of bacteria, and outbreak isolates may qualify as “measurably evolving”, and thus applicable to dating methods, particularly as sequencing and analytical tools continue to evolve more power and accuracy. However, the field of phylogenetic dating in fungal pathogens is still in its infancy with differing strengths and deficits in approaches taken to date. For future phylodynamic studies in fungi, some of the lessons described here can be applied to aid development of a “best practice” roadmap, which can be improved upon over time as new tools and techniques become available. A reasonable starting point is to ensure the following steps are addressed (as described in Figure 4):Identify and remove regions of recombination, including recombination both within and between sub-populations/clades. This can be done using freely available tools, such as Gubbins or ClonalFrameML, and should be conducted first as recombination can hinder detection of a temporal signal in the data.Test data for temporal signal through a combination of methods including root-to-tip regression, data randomisation and improved model fit. Importantly, if the data does not have measurable temporal signals, then dating analyses should not be applied.Consider the biology of the organism and potential rate heterogeneity in the selection of appropriate clock models, particularly if datasets are limited to just clinical outbreak isolates or if common ancestors are far back in time. Tests of model fit between different clock (and demographic) models can have poor performance and should not be solely relied upon [108]. Since most fungal infections are saprophytic, stemming from the environment, inclusion of environmental fungal isolates in a phylogeny may provide better indication of heterogeneity in clock rates. Further research into fungal states of hypermutation and quiescence may shed more light on these variations and how frequent or infrequent they may be.If utilising Bayesian analysis, there should be a biological justification for prior settings, and all should be explicitly shared in publications. This can ensure that models are replicable and would allow for comparison between different models. Since final trees can be affected by sensitivity to certain priors, being explicit with model settings would allow for this to be investigated. Studies could examine the sensitivity of the final tree to clock rate and tree priors to justify (or question) the validity of the final tree.

## Figures and Tables

**Figure 1 jof-07-00661-f001:**
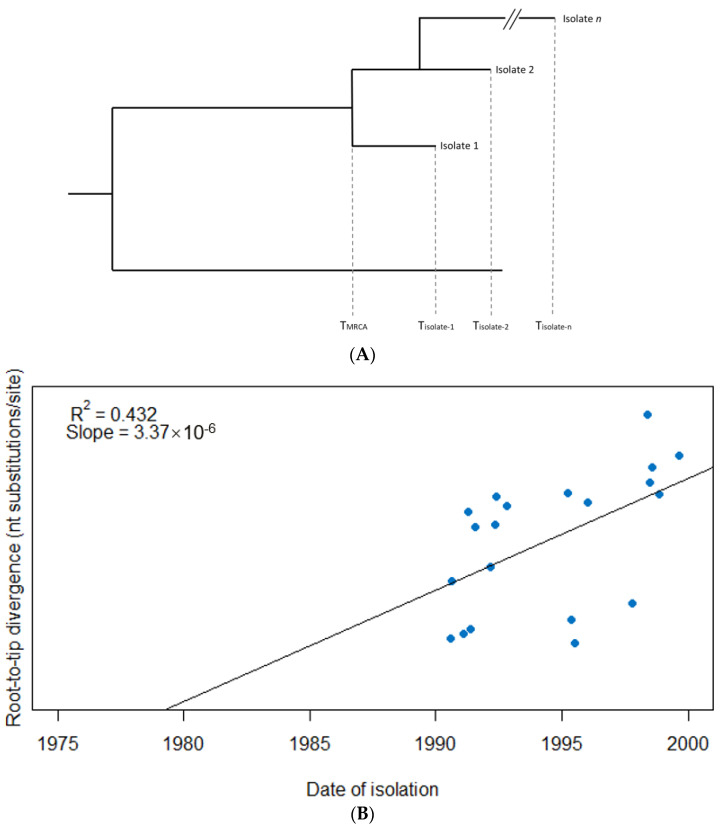
Root-to-tip regression analysis to detect evidence of a temporal signal among 20 hypothetical isolates. (**A**) A phylogenetic tree from which the root-to-tip distance for each isolate is calculated, i.e., the evolutionary change between the time to most recent common ancestor of the sample (TMRCA) and the sampling time of each respective isolate (T_isolate-1_, T_isolate-2_, T_isolate-n_), usually measured as the number of nucleotide substitutions per site over the respective time period. (**B**) The isolates are plotted based on their date of isolation (x-axis) and root-to-tip divergence, i.e., the number of nucleotide substitutions per site (y-axis). The linear regression provides a measure of temporal signal (R^2^) as well as molecular clock rate assuming a strict clock is to be applied (slope). In this example, an R^2^ of 0.432 suggests evidence of a temporal signal in the data, while the (strict) molecular clock rate is estimated at 3.37 *×* 10^−^^6^ subs/site/year.

**Figure 2 jof-07-00661-f002:**
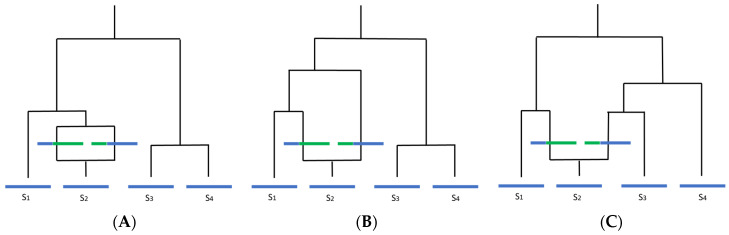
How recombination affects molecular clock-like processes and interpretation of the underlying phylogeny (adapted from Hein et al., 2005). Blue indicates ancestral material, green indicates non-ancestral material. Assuming there is only one recombination event in the history of the sample, to the left of the recombination event, there is one local tree, and to the right, there is another local tree. This can alter the inferred overall phylogeny in three ways: (**A**) looking back in time, if the two recombining sequences merge with each other before coalescing with any other sequence, the trees will be identical and the recombination event will likely be undetectable; (**B**) if one of the recombining sequences coalesces with a difference sequence (in this case, S1) before merging with the other recombining sequence, the unrooted tree topology will remain the same, but branch lengths will change; (**C**) if two or more sequences coalesce with the recombining sequences before the two recombining sequences merge, the tree topology will change.

**Figure 3 jof-07-00661-f003:**
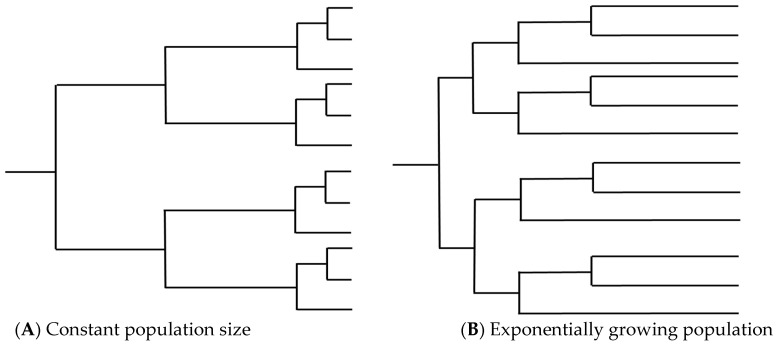
Examples of tree topology from (**A**) a population with constant population size and (**B**) a population undergoing exponential growth. High levels of recombination in a set of isolates can lengthen terminal branches, leading to trees that resemble (**B**) and resulting in incorrect inference relating to population dynamics over time.

**Figure 4 jof-07-00661-f004:**
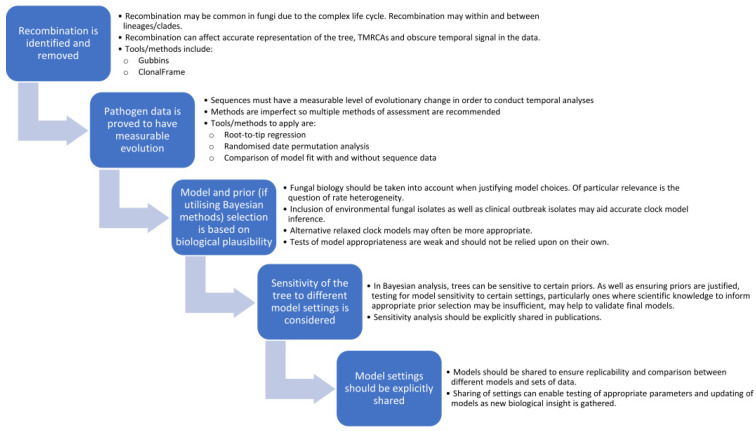
The fungal temporal analysis roadmap—recommendations for future fungal phylogenetic dating studies to ensure biology is taken into account and phylogenies are valid and replicable.

**Table 1 jof-07-00661-t001:** Software tools available for dating phylogenies.

Task	Tool/Algorithm	Description	Reference
Identify loci of recombination	Gubbins	Iteratively identifies loci of recombination and simultaneously constructs a phylogeny based on point mutations outside of these identified regions.	Croucher et al., *Nucleic Acids Res.* 2014, doi:10.1093/nar/gku1196
	ClonalFrameML	Maximum likelihood inference to detect loci of recombination and simultaneously construct a phylogeny accounting for this recombination.	Didelot and Wilson, *PLoS Comput. Biol.* 2015, doi:10.1371/journal.pcbi.10004041
	fastGEAR	Identifies population genetic structure of an (species-wide) alignment and detects recombination, both recent and ancestral, between inferred lineages as well as recent recombination from external origins using a hidden Markov model.	Mostowy et al., *Mol. Biol. Evol.* 2017, doi:10.1093/molbev/msx066
Phylogenetic inference	BEAST/BEAST2	Bayesian MCMC algorithm to jointly estimate a phylogeny and its associated parameters (i.e., effective population size, TMRCA, clock rate, etc.)	Drummond and Rambaut, *BMC Evol. Biol.* 2007, doi:10.1017/CBO9781139095112.007
	MrBayes	Bayesian MCMC inference and model choice across a range of phylogenetic and evolutionary models	Huelsenbeck and Ronquist, *Bioinformatics*. 2001, 17 (8), 754–755
	IQ-Tree	Stochastic tree-searching algorithm to identify the highest likelihood tree (output in nucleotide substitutions only, not calendar time)	Nguyen et al., *Mol. Biol. Evol*. 2015, 32(1):268–74.
	PhyML	ML inference of phylogenetic relationships between divergent populations, utilising subtree pruning and regrafting (SPR) and approximate likelihood-ratio test (aLRT) approaches.	Guindon et al., *Systematic Biol.* 2010, 59 (3), 307–321.
	RAxML	ML inference of phylogenetic relationships between divergent populations utilising parsimony and heuristic subtree rearrangements.	Stamatakis, *Bioinformatics.* 2014, 30 (9), 1312–1313.
	PHYLIP	Package of programmes for phylogenetic inference including parsimony, distance matrix and ML methods, bootstrapping and consensus trees.	Felsenstein, *Cladistics*. 1989, 5 (2), 163–166.
	SNPhylo	Pipeline utilising ML to reconstruct phylogenies based on SNP data.	Lee et al., *BMC Genomics.* 2014, 15, 162.
Molecular clock rate/Divergence time estimation	Treedater	R package to apply an evolutionary timescale to date and root a phylogeny (i.e., transforms branch lengths from number of nucleotide substitutions to calendar time) and estimate TMRCA. Molecular clock test function tests for appropriate clock model (relaxed vs. strict).	Volz and Frost, *Virus Evol*. 2017, doi:10.1093/ve/vex025
	PhyTime	A tool in the PhyML package that estimates divergence dates in a Bayesian setting.	Guindon, *Systematic Biol.* 2013, 62 (1), 22–34 Top of Form Bottom of Form
Phylogeny viewer/editor	Figtree	Graphical viewer of phylogenetic trees and to produce publication-ready figures. Particularly suited to trees produced by BEAST.	http://tree.bio.ed.ac.uk/software/figtree/ accessed on 11 August 2021
	Icytree	A simple browser-based phylogenetic tree viewer.	https://icytree.org/ accessed on 11 August 2021
	GGTREE	An R package for programmable visualisation and annotation of phylogenetic trees.	Guangchuang et al., *Methods in Ecology and Evolution*. 8 (1), 28–36

N.B. This list is by no means exhaustive. Readers can find a more comprehensive database of programs relevant to some of these tasks at http://methodspopgen.com accessed on 11 August 2021.

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
