# Peer review of "Accounting for the Biological Complexity of Pathogenic Fungi in Phylogenetic Dating"

_jof, 2021, doi:10.3390/jof7080661_

Round 1

Reviewer 1 Report

The Manuscript "Accounting for the biological complexity of pathogenic fungi in phylogenetic dating" is a review centered around the temporal dating of phylogenies for pathogenic fungi. The whole pathogenic fungal community will greatly benefit from this review.

This topic is absolutely important in epidemiology and in understanding the virulence attributes and antimycotics resistance of fungal pathogenic species.

The authors are both well known in their respective fields and have a wide understanding of the topic covered in this review, based on their publication history. In fact, this review is something that I myself plan to utilize in my own work to gain insights into phylogenomic dating and genomic epidemiology.

The abstract is very clear and gives a highly interesting overview of several hot topics in pathogen research. The authors lead the reader through the topics precisely, with adequate references and they even call attention on the lack of some techniques in their previous works, so the whole review is very objective and reliable.

A few suggestions:

in lines 53-55, the authors could also call attention on the fact that these few species do not even represent all phyla that have pathogenic members.

Lines 89-91: title could be moved to next page.

Line 155: for a general reader, a short note on what clades are may be useful

Lines 189:

Fungal sexual reproduction can be heterothallic when two opposing mating types come 189 into contact and mate with each other, or homothallic whereby two cells of a single 190 mating type mate with each other [51].

Rare mating also a possibility, it is not necessarily distinguished from homothallic mating, but at least in Saccharoymces literature, it refers to mating between cells of different ploidy. This would further complicate the issue. https://www.ncbi.nlm.nih.gov/pmc/articles/PMC1212722/

Figure 4: maybe turn it on its side!

Author Response

We thank Reviewer 1 for such kind words and positive feedback! We’re delighted that this review will be utilised by yourself; we’re glad it’s been helpful. We have taken their comments into account:

  1. We have added a disclaimer on Lines 55-56 that these four species included in this review are not representative of all fungal phyla with pathogenic members.
  2. Line 89 – title moved onto next page
  3. Line 153-157: we have added clarification on the four auris clades.
  4. Addition of rare mating added in Lines 214-218
  5. Figure 4 has been rotated 90 degrees

Reviewer 2 Report

COMMENTS TO THE MANUSCRIPT “Accounting for the biological complexity of pathogenic fungi in phylogenetic dating” by Edwards and Rhodes.

General comment:

In the submitted manuscript the authors highlight the relevance of constructing dated phylogenies of fungal pathogens in order to better understand its evolutionary forces involved in virulence, drug resistance and outbreak emergences. The manuscript reviews the four published studies in which temporal dating phylogenies has been conducted for the fungal pathogens Cryptococcus gattii, Paracoccidioides spp., and Candida auris. They discuss the strength and weakness regarding the application of dating phylogenies of such studies and compare it with the previous work on dating phylogenies on virus and bacteria. Based in the analysis of such antecedents, the authors propose a roadmap for the adequate use of temporal dating phylogenies in future studies of fungal pathogens.

The submitted manuscript is well written, the methodology is properly described, and the subject is clearly explained and reviewed. By this, the subject of the submitted manuscript if of interest both in fungal evolution and public health areas and is suitable to be published in the Journal of Fungi. Below are some specific comments for the authors' consideration.

Specific comments:

1. The concepts “evolutionary rate” and “mutation rate” are used indistinctly in the text. Despite these concepts are usually indistinctly used in several references, a nucleotide change within the genome not necessarily has an evolutionary consequence, they are neutral. As authors correctly state, the fungal evolution is a consequence of diverse biological and molecular mechanisms that includes horizontal gene transfer (HGT), variations in chromosomal number (ploidy), hybridization, mobile genetic elements, etc. Thus, I suggest to the authors to consistently use “mutation rate” in their work, avoiding the use of “evolutionary rate”.

2. Not only studies on dating phylogenies from outbreak associated pathogenic fungi are scarce, but there are also there are few experimental data of mutational rates of fungi. Among the few experimental mutation rates published for fungal pathogens are the works on Cryptococcus neoformans (Xu, 2004. Genetics. 168(3): 1177-1188) and Aspergillus (Álvarez-Escribano et al. 2019; BMC Biology. 17:88). How can experimental data aid to calibrate molecular clocks for dating phylogenies? Or there is not needed to link such experimental works with data from outbreaks?

3. Commonly, this kind of review is the first contact with the subject for must of the readers. Thus, it would be useful to add a table with the main bioinformatic tools available to conduct a dating phylogeny. BEAST, Gubbins and ClonalFrameML and fastGEAR are named in the manuscript, but it would be useful to add a table with a general description of these tools and maybe other ones.

4. Complementary to the previous suggestion, although the included references are adequate, I recommend adding the references: Rutschmann, 2006. Diversity and Distributions. 12(1): 35-48; and Dos Reis et al. 2016. Nature Reviews Genetics. 17(2): 71-80. These are not case-specific references that review the general methods and bioinformatic tools to generate a dating phylogeny which will be very useful for readers who begin to interest in use these tools.

Author Response

We thank Reviewer 2 for the positive and constructive feedback! Some more detailed responses are below:

  1. We have changed all instances of evolutionary rate to mutation rate throughout.
  2. Whilst this is actually a great idea, the authors think it would need to be carried out on more isolates than tested in these papers. An ideal experiment would include WGS at multiple points in a time course, with replicates, over a wide population. For example, the Alvarez-Escribano paper provides great estimates for Aspergillus species, but we know that the genetic diversity of Aspergillus is incredibly broad; it could be that the rates estimated here are out just simply due to not sampling the diversity space sufficiently. However, this comment is valid – it would just be very expensive to do such an experiment.
  3. A table for the comparison of software has been added on pages 8-10.
  4. The authors respectfully disagree with the suggestion of the addition of Rutschmann 2006, as it is now 15 years old and some of the content is now out of date; however, the reference to Dos Reis is included in Line 381, thank you for this suggestion.

Reviewer 3 Report

This manuscript is well written and worth publishing in Journal of Fungi. I have few minor comments i.e. please arrange key words in alphabetical order, don't repeat words in the title as key words, please make sure n dash is used in between numerals in reference section. 

Author Response

Thank you for your suggestions. We have made the necessary changes by updating the keywords as requested and the reference section.